# Examining the Potential Applicability of Orexigenic and Anorexigenic Peptides in Veterinary Medicine for the Management of Obesity in Companion Animals

**Cezary Osiak-Wicha** [1], **Katarzyna Kras** [1], **Ewa Tomaszewska** [2], **Siemowit Muszyński** [3] and **Marcin B. Arciszewski** [1,*]

[1] Department of Animal Anatomy and Histology, Faculty of Veterinary Medicine, University of Life Sciences in Lublin, Akademicka 12, 20-950 Lublin, Poland; cezary.wicha@up.lublin.pl (C.O.-W.); katarzyna.kras@up.lublin.pl (K.K.)

[2] Department of Physiology, Faculty of Veterinary Medicine, University of Life Sciences in Lublin, Akademicka 12, 20-950 Lublin, Poland; ewarst@interia.pl

[3] Department of Biophysics, Faculty of Veterinary Medicine, University of Life Sciences in Lublin, Akademicka 13, 20-950 Lublin, Poland; siemowit.muszynski@up.lublin.pl

* Correspondence: mb.arciszewski@wp.pl

**Abstract:** This review article comprehensively explores the role of orexigenic and anorexigenic peptides in the management of obesity in companion animals, with a focus on clinical applications. Obesity in domestic animals, particularly dogs and cats, is prevalent, with significant implications for their health and well-being. Factors contributing to obesity include overfeeding, poor-quality diet, lack of physical activity, and genetic predispositions. Despite the seriousness of this condition, it is often underestimated, with societal perceptions sometimes reinforcing unhealthy behaviors. Understanding the regulation of food intake and identifying factors affecting the function of food intake-related proteins are crucial in combating obesity. Dysregulations in these proteins, whether due to genetic mutations, enzymatic dysfunctions, or receptor abnormalities, can have profound health consequences. Molecular biology techniques play a pivotal role in elucidating these mechanisms, offering insights into potential therapeutic interventions. The review categorizes food intake-related proteins into anorexigenic peptides (inhibitors of food intake) and orexigenic peptides (enhancers of food intake). It thoroughly examines current research on regulating energy balance in companion animals, emphasizing the clinical application of various peptides, including ghrelin, phoenixin (PNX), asprosin, glucagon-like peptide 1 (GLP-1), leptin, and nesfatin-1, in veterinary obesity management. This comprehensive review aims to provide valuable insights into the complex interplay between peptides, energy balance regulation, and obesity in companion animals. It underscores the importance of targeted interventions and highlights the potential of peptide-based therapies in improving the health outcomes of obese pets.

**Keywords:** appetite regulation; cats; dogs; veterinary medicine; animal health

## 1. Introduction

Obesity, characterized by improper or excessive accumulation of adipose tissue, is a disease that affects many organs and impacts the well-being of domestic animals [1]. According to the U.S. Pet Obesity Report of the Association for Pet Obesity Prevention (APOP) in 2022, 59% of dogs and 61% of cats in the U.S. were overweight or obese [2].

The occurrence of obesity in companion animals is largely influenced by the owner and other individuals with whom the animal interacts on a daily basis. Overfeeding, poor-quality diet, failure to adjust feeding practices to the animal's health status and activity level, as well as lack of physical activity play significant roles (Figure 1) [1,3]. As a result, this condition is often stigmatized not only in humans but also in pets [4].

However, certain diseases, metabolic disorders (i.e., diabetes mellitus, hypothyroidism, hyperadrenocorticism) or genetic predispositions may also contribute to the development of obesity, although the extent to which heredity influences obesity development is still under investigation [5–7]. Research to date suggests that susceptibility to the obesity development may be hereditary up to 71–81% in humans [8]. This result indicates the need to investigate the intrinsic factors responsible for this disease, which could potentially be applicable in obese humans and also in animals.

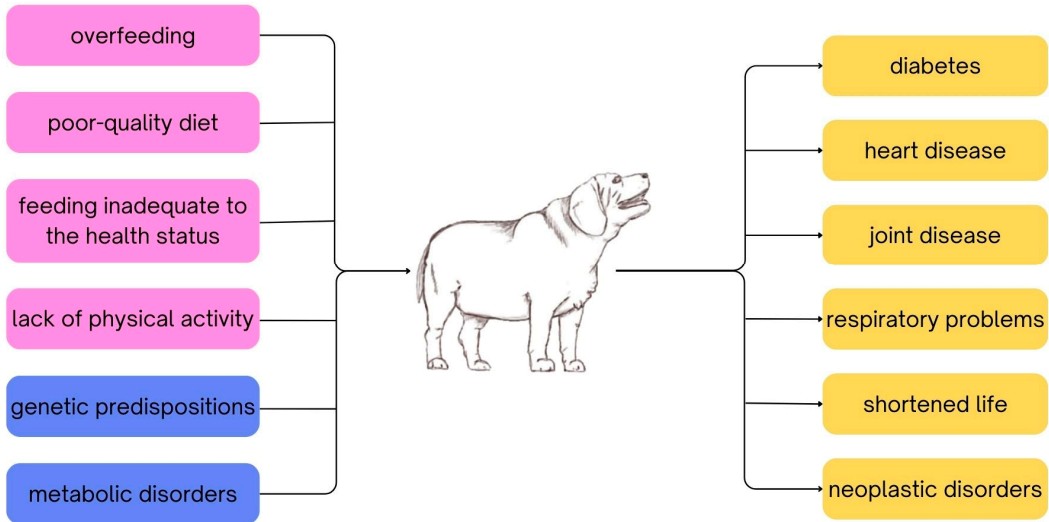

**Figure 1.** Causes (pink and blue boxes) and consequences (yellow boxes) of obesity in companion animals. Pink boxes indicate causes due to human factors; blue boxes indicate intrinsic factors.

Companion animals are a special group of animals, particularly given their diet, which differs from that of other domesticated animals. Unlike farm animals, animals for slaughter or animals bred for other typically utilitarian purposes (animal products), companion animals, particularly dogs and cats, are fed not only to meet their energy needs, but also to play or train, which often leads to the animal's energy needs being exceeded. This in turn can lead to the development of overweight and, in extreme cases, obesity [1]. Furthermore, obese pets, especially cats, are perceived as "funny" and "cute" according to social media, which play a huge role in the modern world, while additionally obese dogs are perceived as "greedy", as shown by the viewing statistics for videos with the relevant hashtags provided by the APOP and an analysis by Lloyd et al. [9,10]. This shows that the problem of obesity in companion animals is underestimated. Excessive body weight can lead to a number of serious health problems, such as heart disease, diabetes, joint disease, respiratory problems, neoplastic disorders (i.e., urogenital cancer, mammary tumors), consequently shortening the life of the animal. It can also lead to a significant deterioration in the animal's quality of life [11,12]. Therefore, combating obesity in companion animals is becoming increasingly important, for the benefit of both the animals and their owners [5]. It is worth paying attention to educating pet owners about proper nutrition, regular physical activity, and monitoring their pets' weight to prevent overweight- and obesity-related problems. However, as already mentioned, the factors causing obesity are diverse and the behavior of the pet owners is not the only one. Undoubtedly, food intake plays an essential role in the development of this disease.

Food intake and absorption are key elements in maintaining the body's energy homeostasis. These processes form a cycle comprising three main stages, namely hunger, satiation and satiety [13]. The regulation of these processes depends on complex mechanisms, including the activity of food intake-related proteins. These proteins play an important role both in the central nervous system and in peripheral organs, including the gastrointestinal tract (GIT) and adipose tissue, where they are secreted [14]. In different circumstances,

abnormalities in the function of these proteins can occur. These abnormalities can have a variety of sources, ranging from mutations in genes encoding the protein through dysfunctions of enzymes necessary for the proteins to function properly to disruptions in the structure or function of the receptors to which the proteins should bind. These subtle dysregulations can lead to serious health consequences, which highlight the importance of understanding the mechanisms that regulate food intake and identifying factors that affect the function of proteins involved in this process [15]. Thus, the use of molecular biology techniques to study and understand mutations in genes, the study of gene expression, post-translational modifications of the proteins themselves but also of the enzymes responsible for their proper function and of the receptors for which proteins are ligands may be the key to understanding and combating obesity. Food intake-related proteins can be generally divided into two main groups: orexigenic peptides (enhancement of food intake) and anorexigenic peptides (inhibition of food intake) [16]. This review aims to thoroughly examine current research on regulating energy balance in companion animals, emphasizing the clinical application of orexigenic and anorexigenic peptides, including ghrelin, phoenixin (PNX), asprosin, glucagon-like peptide 1 (GLP-1), leptin, and nesfatin-1, in veterinary obesity management.

## 2. Materials and Methods

The literature search for this review article on orexigenic and anorexigenic peptides involved comprehensive exploration of multiple scientific databases, including PubMed, Scopus, Web of Science, and Google Scholar. A combination of specific and broad search terms such as "orexigenic peptides", "anorexigenic peptides", "appetite regulation", "ghrelin", "asprosin", "GLP-1", "leptin", "peptide mechanisms", "peptide therapy in obesity", "peptide therapy in diabetes", and "peptides in veterinary medicine" was utilized, with Boolean operators (AND, OR) applied to refine search results.

The inclusion criteria comprised peer-reviewed articles, reviews, and original research papers published in English from 2000 to 2024 (with the exception of articles where the peptides were first mentioned, or with lack of newer articles on the subject), focusing on the biochemical and physiological aspects of orexigenic and anorexigenic peptides, their therapeutic implications, and related mechanisms. Studies specifically addressing the effects of these peptides on appetite regulation, obesity, diabetes, and metabolic syndromes were prioritized. Exclusion criteria included articles not available in English, studies lacking a clear focus on orexigenic or anorexigenic peptides or their mechanisms, non-peer-reviewed sources unless providing critical insights, and redundant studies or those with insufficient data.

The initial search yielded a substantial number of articles, which were screened based on titles and abstracts to filter out irrelevant studies. Full-text articles of potentially relevant studies were retrieved and further assessed for inclusion based on the predefined criteria. The filter settings involved selecting articles published in English, focusing on the specified time frame, and prioritizing studies with a clear emphasis on orexigenic and anorexigenic peptides and their therapeutic implications.

The number of results obtained during the search varied across databases, with PubMed yielding the highest number, followed by Scopus, Web of Science, and Google Scholar. After applying the filters and screening processes, a final set of relevant articles was selected for detailed analysis and synthesis of key information. Special attention was given to studies involving animal models, particularly those relevant to cats and dogs, to explore veterinary implications comprehensively.

## 3. Orexigenic Peptides

Orexigenic peptides, among them ghrelin, PNX and asprosin, are a class of signalling molecules within the body that play a fundamental role in regulating appetite and energy balance [17,18]. Derived from various sources such as the GIT, hypothalamus and adipose tissue, these peptides exert their effects by interacting with specific receptors in

the central nervous system and peripheral organs. One of their functions is to stimulate appetite, promoting food intake and ensuring adequate energy supply for physiological processes [16,19–21]. Orexigenic peptides also participate in the intricate network of signals that govern hunger, satiety, and metabolic responses to nutrient availability. Through their actions, orexigenic peptides contribute to the maintenance of energy homeostasis, orchestrating a delicate balance between food intake and energy expenditure essential for overall health and survival.

*3.1. Ghrelin*

Ghrelin, a remarkable peptide hormone composed of 28 amino acids, stands as a pivotal regulator within the intricate network of physiological processes governing appetite, energy balance, and growth hormone secretion. Its discovery in the late 1990s sparked a revolution in endocrinology, unveiling novel insights into hunger and satiety mechanisms that profoundly impact human health and well-being [22]. Ghrelin undergoes a complex process from gene transcription to peptide synthesis and posttranslational modification. The human ghrelin gene, located on chromosome 3, consists of six exons, encoding a 511 bp mRNA [23,24]. Upon transcription, the ghrelin gene generates preproghrelin, a precursor peptide comprising a 23-amino-acid signal peptide and a 94-amino-acid proghrelin segment. Proghrelin is further cleaved to yield the 28-amino-acid ghrelin peptide and a 66-amino-acid carboxyterminal peptide called C-ghrelin. Additionally, alternative splicing of the ghrelin gene produces transcripts encoding other peptides, including des-Gln14-ghrelin. Interestingly, antisense transcripts may also generate noncoding RNAs, potentially involved in posttranscriptional and posttranslational gene regulation. The enzymes responsible for processing preproghrelin into ghrelin include signal peptidase, prohormone convertase 1/3 (PC 1/3), and a carboxypeptidase-B like enzyme. Signal peptidase cleaves at Arg23, while PC 1/3 cleaves at Arg51, generating ghrelin 1–28 [25]. Subsequently, a carboxypeptidase-B-like enzyme cleaves at Pro50, yielding ghrelin 1–27. Although the involvement of prohormone convertase 2 (PC2) and furin remains controversial, these enzymes may contribute to the processing of preproghrelin [25,26]. After peptide synthesis, ghrelin undergoes a unique posttranslational modification known as acylation. This process involves the addition of an acyl group, typically an octanoyl group (C8:0), to the hydroxyl group of the Ser3 residue. Ghrelin can also be acylated by decanoyl (C10:0) or decenoyl (C10:1) groups, albeit less commonly. Ghrelin acylation is mediated by the enzyme ghrelin O-acyl transferase (GOAT), which belongs to the membrane O-acyl transferases (MBOAT) family [27,28]. GOAT acylates ghrelin with fatty acids ranging from C:7 to C:12, with C:8 being the most prevalent. This acylation process occurs in the endoplasmic reticulum, prior to the processing of proghrelin by various proteases. Unfortunately, information on the specific number of exons and mRNA size in the dog and cat ghrelin gene is currently limited. Notably, the amino acid sequence of ghrelin exhibits remarkable conservation across species, with a high degree of similarity observed among humans, pigs, cows, sheep, cats, and dogs [29–31]. Unlike many gastrointestinal hormones secreted into the lumen of the GIT, ghrelin is released into the bloodstream for systemic distribution. Its secretion is tightly regulated by food intake, with levels increasing during fasting periods and decreasing after food consumption. This regulatory mechanism underscores ghrelin's role in appetite control and energy homeostasis across various mammalian species, as demonstrated in studies involving dogs and cats [27,32]. Ghrelin exerts its effects primarily through the activation of a specific G-protein-coupled receptor known as the growth hormone secretagogue receptor 1a (GHS-R1a). This receptor is widely expressed in areas of the hypothalamus associated with feeding behavior, including the arcuate and ventromedial nuclei. Additionally, GHS-R1a is found in other organs such as the pituitary gland, bone, heart, lung, liver, kidney, pancreas, and immune cells [27]. Research indicates that GHS-R1a can modulate dopamine signaling, potentially through the formation of heterodimers with dopamine receptors in neurons. This interaction has implications for dopamine reward signaling and may influence appetite regulation. Furthermore, studies suggest that

GHS-R1a can form heterodimers with the serotonin 2C receptor, suggesting a role in the regulation of food reward and appetite control. These receptor interactions highlight the complexity of ghrelin's mechanisms of action and its involvement in multiple physiological processes beyond appetite regulation [27]. In addition to its role in stimulating growth hormone (GH) secretion and regulating appetite, ghrelin has garnered attention for its potential therapeutic applications in veterinary medicine. Studies in dogs have revealed that ghrelin administration promptly increases circulating GH concentrations, albeit transiently, highlighting its potency as a GH secretagogue in canines. Furthermore, the distribution of ghrelin-immunoreactive cells in the canine stomach mirrors that of other mammals, with high concentrations observed in the stomach fundus and body. This anatomical localization underscores the evolutionary conservation of ghrelin's physiological role across species. Moreover, the observed increase in daily food intake following ghrelin administration in dogs suggests its direct involvement in the regulation of feeding behavior and energy homeostasis in canines, further emphasizing its potential therapeutic relevance in managing appetite-related disorders [33].

### 3.2. PNX

PNX, a recently discovered (2013) and highly conserved secreted peptide, exists in two primary isoforms: PNX-14 and PNX-20, distinguished by their respective lengths of 14 and 20 amino acids [20]. Initial investigations revealed PNX's crucial role in maintaining normal reproductive function by directly stimulating the release of gonadotropin-releasing hormone (GnRH) from the hypothalamus. GnRH, acting as a pivotal regulator, stimulates the anterior pituitary gland to release follicle-stimulating hormone (FSH) and luteinizing hormone (LH), ultimately orchestrating various reproductive processes [34]. PNX originates from the C-terminal end of a protein called small integral membrane protein 20 (SMIM20), encoded by the *SMIM20* gene. While PC enzymes are suspected to be involved in its cleavage, the exact mechanism remains unclear. Ectodomain shedding, another potential cleavage pathway, also warrants further investigation [17,34]. SMIM20 itself functions as a chaperone-like protein within the mitochondria, stabilizing a crucial subunit of cytochrome c oxidase, an essential component of the electron transport chain. Disruptions in SMIM20 levels can significantly impact cytochrome c oxidase assembly, highlighting its vital role [35,36]. Understanding factors influencing *SMIM20* expression becomes crucial as it directly impacts PNX production. PNX-14 and PNX-20 exhibit distinct localization patterns across various organs. The hypothalamus predominates in the expression of PNX-20, whereas PNX-14 exhibits its highest abundance in the heart and spinal cord [34,37]. Both isoforms require C-terminal amidation for biological activity [20]. Utilizing specific immunoassays, researchers have identified the hypothalamus, particularly the paraventricular and supraoptic nuclei, as the region boasting the highest PNX concentration. PNX is also located in numerous other hypothalamic regions, partially overlapping with nesfatin-1, a peptide with diverse functions [38]. Its journey extends beyond the hypothalamus, reaching the nucleus of the solitary tract, substantia nigra reticulata, dorsal motor nucleus of the vagus, area postrema, and the spinal cord [34,39]. The anterior and posterior pituitary gland, along with the median eminence, also express PNX [20]. Additionally, PNX-14 has been identified in the crypts of the small intestine and the endocrine pancreas [37]. While studies have reported PNX presence in various other organs like the stomach, esophagus, spleen, and kidneys, these findings require further verification. Beyond its established role in reproduction, PNX appears to hold broader significance in various physiological processes. PNX administration in rats has been shown to increase food intake, suggesting an orexigenic effect, however, its role in energy balance is more nuanced [40]. By ensuring adequate intake to meet metabolic demands, PNX-14 may also play a regulatory role in preventing overeating through its interactions with other appetite-suppressing signals when energy stores are sufficient. However, further investigation is needed to elucidate the exact mechanisms and long-term effects of PNX on feeding behavior. PNX's presence in the heart and its ability to modulate GnRH release, which, in turn, can exert cardiovascular

effects, suggest a potential role in regulating blood pressure and heart function. However, more research is required to definitively establish this connection. PNX expression in brain regions associated with memory and learning, coupled with studies showing improved memory performance in animal models upon PNX administration, warrant further research to understand its potential role in cognitive function [41]. Similarly, PNX's presence in brain regions associated with stress and anxiety, along with its ability to modulate stress hormones, has led researchers to explore its potential role in regulating these emotional states. Studies have shown that PNX administration can reduce anxiety-like behaviors in animal models [42]. While the exact mechanisms by which PNX exerts its diverse effects are still being unraveled, a key player in the signaling pathway is the G protein-coupled receptor (GPCR) known as GPR173. This receptor was initially identified as a potential candidate for binding PNX due to its "orphan" status, meaning it lacked a known ligand. GPR173, also referred to as Superconserved Receptor Expressed in Brain 3 (SREB3), is primarily expressed in the brain and ovaries, aligning with PNX's known functions in reproduction, anxiety regulation, and appetite stimulation. Interestingly, its presence is particularly prominent in brain regions like the piriform cortex, lateral septum, and hypothalamus, further suggesting a potential link to PNX's diverse central nervous system actions. Studies investigating the interaction between PNX and GPR173 provide intriguing evidence, but not definitive proof. Knockdown of *GPR173* using siRNA technology has been shown to diminish the effects of PNX on kisspeptin and GnRH regulation, suggesting a potential role for GPR173 in mediating these functions. Additionally, PNX administration has been observed to increase *GPR173* expression, possibly as part of a feedback loop to regulate its own activity [43–45]. If GPR173 is indeed the receptor for PNX, the current understanding suggests it activates the cAMP-protein kinase A (PKA) pathway upon ligand binding. This pathway ultimately leads to the phosphorylation of the transcription factor CREB, which plays a crucial role in regulating gene expression. However, it is important to note that other signaling pathways, such as MAPK and PKC, might not be involved in all of PNX's actions. While the evidence linking PNX to GPR173 is compelling, it is crucial to acknowledge emerging research questioning this relationship. One study points to inconsistencies in the evolutionary distribution of GPR173 compared to PNX, suggesting the possibility of a different receptor being involved [46].

*3.3. Asprosin*

Asprosin, a recently discovered adipokine, is a critical regulator of appetite and glucose metabolism [47,48]. This peptide is primarily secreted by the white adipose tissue (WAT) and the liver, functioning as a C-terminal cleavage product of profibrillin. Its discovery stemmed from investigations into neonatal progeroid syndrome (NPS), a rare genetic disorder associated with *FBN1* mutations, leading to aberrant asprosin processing and secretion [49]. Subsequent research revealed asprosin's broader role in metabolic physiology, particularly in modulating feeding behavior and glucose homeostasis [50]. Asprosin exerts its effects by binding to target cells and initiating signaling cascades. Its orexigenic effects involve binding to orexigenic neurons in the hypothalamus, activating downstream signaling pathways that stimulate appetite and food intake. Additionally, asprosin acts on hepatocytes, activating the cAMP-PKA pathway and promoting hepatic glucose release through gluconeogenesis and glycogenolysis, consequently elevating blood glucose levels. The mechanism of action of asprosin unfolds through a series of intricate steps, beginning with the cleavage of profibrillin by the calcium-dependent furin/PACE serine endoprotease, a member of the subtilisin family [51]. This cleavage event occurs at the R-C-K/R-R motif in the C-terminal domain of profibrillin and is crucial for the subsequent incorporation of fibrillin-1 into the extracellular matrix. While the exact cellular location of profibrillin cleavage remains largely unknown, it is speculated to occur between the trans-Golgi network and the cell surface or upon fibrillin-1 secretion [52]. Asprosin generated in this way is predominantly secreted from WAT, although evidence suggests its secretion from other organs like the skin, pancreas (more specifically, β-cells), and

salivary glands [53]. Notably, asprosin secretion exhibits a circadian rhythm, with levels peaking after overnight fasting and decreasing upon feeding. Conditions of overnutrition, such as obesity and type 2 diabetes, are associated with elevated circulating levels of asprosin, indicating its role in metabolic dysregulation. Upon entering the bloodstream, asprosin targets both the liver and the brain. In the liver, asprosin stimulates hepatic glucose release through the activation of the cAMP-PKA pathway. This glucogenic effect is mediated by a GPCR called OR4M1, which is expressed in the liver [54]. Asprosin binding to OR4M1 triggers intracellular signaling cascades that lead to increased cAMP levels, PKA activity, and subsequent glucose production in hepatocytes. Importantly, asprosin's glucogenic effect is independent of insulin levels or sensitivity, indicating a direct action on hepatic glucose metabolism. Furthermore, asprosin can cross the blood–brain barrier and act in the brain, influencing appetite regulation. It targets orexigenic neurons in the hypothalamus, modulating the activity of Agouti-related peptide (AgRP) and pro-opiomelanocortin (POMC) neurons. By stimulating appetite and food intake, asprosin contributes to the regulation of the energy balance [55]. Though significant strides have been made in understanding asprosin, many facets of its mechanism remain unclear. One notable gap in understanding pertains to the secretion mechanism of asprosin, particularly how WAT and other asprosin-producing organs respond to environmental stimuli and nutritional states to release asprosin. Furthermore, the connection between asprosin's actions in the brain and liver remains puzzling. It is speculated that asprosin might act as a link between the brain's feeding circuitry and the liver's glucose release, possibly by influencing the nervous system's activity in these organs. Despite observations showing that asprosin treatment does not affect certain hormone levels in mice, there could be other central mechanisms at play. While the liver receptor for asprosin is known, the identity of the brain receptor and its relationship with the liver receptor remain unclear.

## 4. Anorexigenic Peptides

Anorexigenic peptides are specific proteins that play a key role in regulating appetite and food intake. They act by inhibiting the feeling of hunger, leading to a reduction in calorie intake. Anorexigenic peptides belonging to hormones, such as GLP-1, leptin and nesfatin-1, send signals to the central nervous system (CNS) [16]. The hypothalamus processes neural and hormonal signals from the GIT and other peripheral organs, synthesizing the data to regulate food intake behavior. Within the arcuate nucleus (ARC) of the hypothalamus, POMC and cocaine- and amphetamine-regulated transcript (CART) neurons play a key role in inhibiting food intake. These neurons are stimulated by excess energy and act to reduce food intake [13]. These peptides therefore play an important role in maintaining energy balance and may be key in the treatment of obesity and other disorders associated with excessive appetite.

### 4.1. GLP-1

Research on glucagon began over a century ago (in the 1920s), and it is now known that glucagon is derived from a larger precursor protein, proglucagon, encoded by the *GCG* gene. Post-translational processing of proglucagon produces various smaller proteins, including glicentin, glicentin-related pancreatic polypeptide (GRPP), glucagon, oxyntomodulin (OXM), the major proglucagon fragment (MPGF), GLP-1, and GLP-2. GLP-1 in particular has garnered significant interest from researchers for its potential in controlling food intake and treating type 2 diabetes [56,57]. The expression of *GCG* is regulated by a single promoter, which, along with controlling sequences, is located within the 2.5 kb 5′-flanking region of the transcription start site in rodents. Expression initiation in the pancreas, brain, and intestines occurs at the same start codon. The entire 2.5 kb region is necessary for transcription in the intestines, whereas a 1.3 kb region is sufficient for transcription in the pancreas and brain, contributing to tissue-specific expression. Additional control of tissue-specific *GCG* expression is provided by proteins binding to the promoter (which contains at least five cis-acting elements, G1-G5, and a cAMP response element

(CRE) in rats) and/or the enhancer region [56]. The transcription factor paired box protein 6 (Pax6) appears to be essential, as mice with a dominant-negative mutation in this gene do not exhibit *GCG* expression in the small and large intestines, nor do they show GLP-1 immunoreactivity. Studies on intestinal cell cultures have shown that the activation of protein kinase A (PKA) by cAMP, independent of the CRE, also regulates *GCG* expression. Factors that enhance *GCG* expression in the intestine include lithium, β-catenin, protein hydrolysates, and insulin. The latter is particularly interesting because insulin inhibits glucagon production in pancreatic islets [56,57]. Pancreatic and duodenal homeobox 1 (PDX1) is a factor that inhibits *GCG* expression, but it requires cooperation with other transcription factors. A newly discovered regulator of *GCG* transcription and inducer of GLP-1 synthesis in intestinal L cells is microRNA miR-194 [58]. Although high *GCG* expression occurs in three main sites—the pancreas, the nucleus tractus solitarius (NTS) in the brainstem, and enteroendocrine L cells in the intestine—GLP-1 is primarily produced in the enteroendocrine L cells and the NTS. This is because proglucagon is post-translationally cleaved at different sites by tissue-specific PCs, leading to the tissue-specific presence of various proglucagon-derived peptides. GLP-1 is generated by the enzymatic cleavage of proglucagon by PC1/3, similar to the processing of the orexigenic peptide ghrelin and the anorexigenic peptide nesfatin-1. This enzyme is also associated with obesity in humans [59]. GLP-1 belongs to the incretins, which are gut hormones secreted in response to nutrients, increasing insulin secretion in the β-cells of the pancreatic islets [60]. In this case, the main trigger in the majority of species is the presence of glucose in the intestinal lumen, primarily in the ileum. In dogs, in contrast, the main trigger is fat diet, and secretion occurs mainly in the jejunum. Additionally, this process in dogs is partially stimulated by another incretin released by K cells in the intestine, which is glucose-dependent insulinotropic polypeptide (GIP). In cats, amino-acid-rich meal results in the highest intestinal GLP-1 release [61]. Due to the rapid increase in serum GLP-1 levels after a meal, it is believed that other mechanisms such as neural conduction and the involvement of taste receptors also stimulate hormone secretion [62]. The primary actions of GLP-1 include stimulating insulin secretion, inhibiting glucagon secretion (which is interesting given that GLP-1 and glucagon originate from the same precursor protein), and reducing appetite while increasing the body's energy expenditure. These effects can be used to treat obesity, which is already being applied in humans. The mechanism by which GLP-1 suppresses appetite is not entirely clear, given that this hormone is rapidly inactivated by the enzyme dipeptidyl peptidase-4 (DPP-4) [63]. The resulting metabolite is considered inactive and may act as a competitive GLP-1 receptor antagonist, but this has not been confirmed in vivo. The latest reports indicate that the GLP-1 metabolite, GLP-1(9-36), inhibits glucagon secretion, thus making it a biologically active compound, contrary to the previous claim of inactivity [64]. The initial form of GLP-1 has an estimated duration of action below 10 min, which is not long enough for the active hormone to leave the intestines. However, studies suggest that an average of 10–15% of the hormone produced in L cells can exit the intestine in its active form [57,65]. It is speculated that GLP-1 might function in the intestine primarily as part of the ileal brake mechanism, which helps control food intake. However, the presence of GLP-1 receptor (GLP-1R) in locations far from L cells such as in pancreatic cells suggests a paracrine activity of GLP-1, despite its rapid degradation [66]. Intestinal-derived GLP-1 might also act through the vagus nerve, which innervates the small intestine, as many gut-derived peptides involved in food intake regulation function this way [67]. It was suspected that GLP-1 signals a reduction in food intake via glutamatergic signaling from the vagus nerve to the NTS. However, a study by Brierley et al. in 2021 found that while NTS neurons do receive signals from the GIT through the vagus nerve, this mechanism involves the activation of oxytocin receptors on vagal afferent neurons (VANs), not GLP-1R [68]. It is important to note that GLP-1 is also produced in the NTS. Brierley et al.'s study found that the NTS does play a role in inhibiting food intake, but primarily in response to very large meals. Although GLP-1 receptors are present on vagal afferent neurons (VANs), the study suggests that the stimulation of these VANs is more motor-sensory rather than chemosensory, as might be

expected given the presence of the receptors. Thus, the study concluded that peripheral and central GLP-1 mechanisms operate independently [68]. Overall, studies indicate that the suppression of food intake mediated by GLP-1 occurs through neurons expressing GLP1R. GLP-1R is part of the large family of G-protein-coupled receptors and is widely distributed in brain structures. One such location is the hypothalamus, specifically the ARC, a well-known center for integrating hunger and satiety signals [69]. It is known that the hypothalamus also transmits signals to the mesolimbic reward system (MRS), a key structure responsible for addiction, impulsive behavior, and food rewards. Alhadeff et al. demonstrated GLP-1R expression in the MRS and further showed that this area participates in the control of food intake [70]. Their experiment first revealed that neurons in the NTS project directly to the MRS. In addition, they found that activation of GLP-1R in this region reduces food intake and body weight, while blocking GLP-1R in the MRS leads to increased food intake and weight gain [70]. In subsequent years, Hsu et al. found activation of GLP-1R in the ventral hippocampal formation (HPFv), a region identified as another site involved in the regulation of food intake. In their experiment, rats were administered a receptor agonist and antagonist, and receptor activity in this region was observed. The agonist led to reduced food intake and weight loss, while the antagonist resulted in increased food intake. Furthermore, they did not detect GLP-1R immunoexpression in axon terminals in the HPFv, which might suggest a mechanism of action based on volume transmission [71]. Another site in the CNS expressing GLP-1R is the lateral dorsal tegmental nucleus (LDTg) of the mesopontine tegmentum. The LDTg is associated with effects on mesolimbic neurotransmission and the basal ganglia of the brain, influencing behaviors such as motivated actions and psychostimulant-induced locomotion. In their study, Reiner et al. demonstrated that activation of GLP-1R with an agonist in the LDTg reduces food intake without causing nausea or malaise in rats [72].

### *4.2. Leptin*

The history of leptin begins with the identification of the gene responsible for obesity, the obese gene, *LEP*, which encodes 16-kDa leptin [73]. Leptin secretion is stimulated mainly by insulin, steroid hormones, and norepinephrine. In contrast, the sympathetic nervous system inhibits leptin secretion through activation of adrenergic receptors. Because the rate of leptin release depends primarily on the rate of *LEP* gene transcription and translation, leptin levels in the circulation are rather stable in the short term and require more time than the previously described GLP-1 to respond to various metabolic stimuli [74]. Among the factors controlling leptin expression is the distant enhancer sequence of leptin 1 (LE1) located 16 kb upstream of the transcription start site (TSS) of the leptin gene. LE1 binds to the peroxisome proliferator-activated receptor gamma (PPARγ)/retinoid X alpha (RXRα) receptor, also known as the leptin regulatory element 1 (LepRE1), which is essential for fat-regulated expression. There is also a functionally analogous site for LepRE1 in another DNA regulatory element, 13 kb downstream of the *LEP* gene TSS. Epigenetic control is also considered, as promoter methylation of the *LEP* gene has been observed in obese patients [75,76]. Leptin is a hormone primarily produced by adipocytes and plays a crucial role in regulating energy balance and body weight, acting as an anorexigenic peptide [77]. A positive correlation between body mass and serum leptin levels was confirmed in both dogs and cats [78–80]. Herein lies a paradox, for as leptin is produced by adipocytes, a decrease in appetite and thus weight loss should be observed in obese individuals; however, this is not the case. This phenomenon is called leptin resistance and its basis may be, among other things, the reduced access of the hormone to its receptor due to changes in receptor expression or changes in signal transduction downstream of the receptor, Lep-R or Ob-R. As a result, the body's ability to regulate the energy balance is disrupted. This can lead to increased appetite, reduced energy expenditure, and difficulties in losing weight [77,81]. Leptin resistance makes simple obesity treatment in the form of leptin administration ineffective [74]. Beyond appetite regulation and general energy balance, leptin also influences various physiological processes, reproductive function,

immune response, and bone health. Moreover, leptin levels can be influenced by factors such as sleep, stress, and inflammation [77]. The exact mechanisms of action of leptin are still unclear, but studies to date indicate that leptin reaches the brain via direct transport across circumventricular organs, saturated transport across the blood–brain barrier, and uptake into the brain parenchyma and choroid plexus [74]. Leptin acts via the Lep-R or Ob-R receptor belonging to the class I cytokine receptor family [82]. Lep-R exists in six isoforms which are produced by alternative RNA splicing of the *LEPR* gene. The receptor isoforms differ in structure and are divided into three classes: long (Lep-Rb), short (Lep-Ra, Lep-Rc, Lep-Rd, Lep-Rf), and secretory (Lep-Re). Lep-Ra, the short form of the receptor, plays an important role as it is involved in the transport of leptin across the blood–brain barrier (BBB) [82,83]. This receptor probably cooperates with specialized hypothalamic glia—tanycytes [84]. However, the study by Yoo et al. contradicts this statement, because they do not demonstrate a connection between tanycytes and the leptin receptor [85]. Among all isoforms, only the long form, Lep-Rb, is able to activate the JAK/STAT pathway and efficiently transmit the signal to the cell. Its expression is highest in the brain, especially in the ARC, ventromedial (VMH), and dorsomedial (DMH) nuclei of the hypothalamus. In these areas, leptin binds to the receptor and, through POMC/CART and Agouti-related peptide/neuropeptide Y (AgRP/NPY) neurons, regulates appetite and energy balance mainly through the activation of signal transducer and activator of transcription-3 (STAT3) [83]. A study conducted by Vong et al. in 2011 showed that the main anti-obesity effects of leptin likely involve acting on presynaptic GABAergic neurons to reduce the inhibitory tone on postsynaptic POMC neurons, leading to increased activity of these neurons. POMC product $\alpha$-MSH activates neurons by binding to the melanocortin receptor (MCR) and leads to appetite suppression [75,86]. In contrast to that study, Xu et al. discovered in 2018 that it is AgRP neurons, not POMC, that are responsible for the main integration of leptin action [87]. Binding of leptin to its receptor also activates the phosphoinositide-3-kinase (PI3K) and the mitogen-activated protein kinase/extracellular signal-regulated kinase (MAPK/ERK) signaling cascades. Together, the Jak/STAT3, MAPK, and PI3K pathways form a complex system controlling the energy balance of the organism [75]. Studies conducted on cats indicate that leptin also acts via the vagus nerve [88]. Leptin is removed from the circulation by glomerular filtration in the kidneys and is then taken up by the tubules and metabolized via the endocytic megalin receptor. These processes are less effective in mice with diet-induced obesity (DIO), leading to hyperleptinemia. In turn, hyperleptinemia may contribute to the development of chronic kidney disease (CKD), which further impairs leptin excretion and exacerbates elevated leptin levels through a harmful positive feedback loop [76,89]. Furthermore, it has been shown that in obese individuals, leptin signaling is disrupted due to endoplasmic reticulum (ER) stress in neurons expressing the leptin receptor. ER stress inhibits leptin signaling through the action of non-receptor protein tyrosine phosphatase type 1 (PTPN1, also known as PTP1B), which dephosphorylates the leptin-activated JAK2 protein [76,90].

### *4.3. Nesfatin*

Nesfatin-1 is a relatively new peptide, discovered in 2006 in rat hypothalamus, and as such, the mechanisms of its action remain to be fully discovered [91]. Nesfatin-1 (N-terminal nesfatin-$1_{1-82}$) is derived from the precursor protein nucleobindin 2 (NUCB2) through post-translational cleavage by hormone PC 1/3 (similarly to ghrelin and GLP-1) as one of the three isoforms, alongside nesfatin-$2_{85-163}$ and nesfatin-$3_{166-396}$. At this moment, it is not possible to specifically target nesfatin-1 in immunohistochemical studies using antibodies, which is why most available research refers to the precursor protein NUCB2 [92]. Expression of *NUCB2* at the gene and protein levels has been demonstrated in various regions of the central nervous system, particularly in areas responsible for food intake control (hypothalamic nuclei) and stress response [93]. Additionally, this peptide is prevalent in the digestive, reproductive, circulatory, and respiratory systems [92,94]. Immunostaining of NUCB2/nesfatin-1 revealed the localization of nesfatin-1 in the cyto-

plasm and primary dendrites, with no presence detected in axons and nerve terminals, which differs from other neuropeptides and suggests an intracellular mode of action for nesfatin-1 [93]. Immunoelectron micrographs indicate that nesfatin-1 localizes mainly in the secretory vesicles of PVN neurons in perikarya near the Golgi apparatus [95]. The anorexigenic mechanism of nesfatin-1 has not yet been fully described. Additionally, the receptor through which nesfatin-1 might act is unknown, although there are some predictions that it could be the ghrelin receptor, GHS-R1a [96,97]. Generally speaking, ghrelin and nesfatin-1 are referred to as sister peptides by some researchers due to their potential shared receptor, colocalization, and the common enzyme that leads to their formation from the precursor protein, which is interesting since ghrelin is an orexigenic hormone, while nesfatin-1 is anorexigenic [98]. Among the mechanisms of action at the CNS level, activation of the anorexigenic CRF 2 receptor [99] and inhibition of orexigenic pathways such as NPY signaling [92] are noted. Maejima et al. demonstrated that nesfatin-1 activates the PVN and NTS areas. In the PVN, nesfatin-1 activates oxytocin neurons and nesfatin-1 neurons themselves, thereby stimulating oxytocin release. Moreover, blocking endogenous nesfatin-1 inhibits oxytocin release in the PVN, suggesting a paracrine/autocrine action of nesfatin-1. In general, studies show that the anorexigenic effect of nesfatin-1 is mediated by oxytocinergic signaling via the melanocortin pathway [95,100]. Colocalization studies of nesfatin-1 and glucocorticoid receptors (GR), known as stress receptors, suggest that stress may influence nesfatin-1 neurons through GR and thus inhibit food intake [101]. In the hypothalamus of obese animals, it was observed that the anti-obesity effect was achieved by nesfatin-1 mediating ERK-dependent sympathetic excitation [102], and in vitro study showed that nesfatin-1, via its receptor, induces CREB phosphorylation, thereby activating an intracellular signaling cascade in neurons [103]. Another study indicates that the regulation of NUCB2, and consequently feeding behavior, may be influenced by the activation of serotonin 5HT2C receptors [100,104]. Furthermore, the anorexigenic effect of nesfatin-1 is potentially associated with histaminergic signaling [100,105]. Interestingly, a 2020 study showed that nesfatin-1 likely acts within the midbrain dopaminergic system, negatively modulating the activity of dopaminergic neurons and thus causing a decrease in hedonic food intake, ultimately leading to an anorexigenic effect [106]. An interesting aspect, seemingly unrelated to food intake, is the correlation of nesfatin-1, oxytocin, serotonin, and dopamine in the serum of aggressive dogs. The relationship between these peptides was demonstrated in a study conducted on pit bulls, in which the authors suggest a possible interaction of these compounds at the level of the central nervous system [107]. This is particularly interesting, because previously mentioned studies have indicated a connection between nesfatin-1 and oxytocin, serotonin, and dopamine. The study on aggressive dogs may suggest that the anorexigenic effect of nesfatin-1 could be due to its influence on the individual's behavior.

## 5. Therapeutic Application

Ghrelin, an orexigenic peptide hormone known for its role in stimulating appetite and food intake, presents significant therapeutic potential in managing obesity in cats and dogs. Ghrelin's ability to increase hunger and meal initiation can be leveraged to establish more consistent and controlled feeding schedules, reducing instances of binge eating and irregular feeding patterns, which are common in obese pets [108]. Incorporating ghrelin or ghrelin receptor agonists into a therapeutic regimen could also help enhance the effectiveness of voluntary exercise programs (Figure 2). As demonstrated in studies with rodents, ghrelin's influence on the central hedonic dopamine system boosts motivation and reward-driven physical activity [108]. This implies that ghrelin administration could make exercise more appealing and rewarding for obese cats and dogs, thereby increasing their physical activity levels [109]. Enhanced exercise contributes to better weight management and overall health, promoting a healthier energy balance and reducing fat accumulation [110]. Furthermore, ghrelin's regulatory effects on circadian rhythms and food anticipatory activity suggest that it can help normalize disrupted daily patterns of

locomotor activity and feeding behavior in obese animals. By restoring a natural daily rhythm, ghrelin can mitigate the effects of obesity on metabolic and behavioral functions, leading to more effective long-term weight control [108]. In clinical settings, ghrelin therapy could be particularly beneficial for pets with obesity-related metabolic disorders. For instance, in cases where obesity is compounded by conditions such as CKD, the manipulation of orexigenic signals through ghrelin could counteract the anorexigenic effects often seen in these diseases [111]. This dual benefit of improving appetite while encouraging physical activity could significantly enhance the quality of life and treatment outcomes for affected animals. In veterinary medicine, the FDA-approved ghrelin receptor agonist, capromorelin (ENTYCE®), has demonstrated safety and effectiveness in stimulating appetite in dogs. Capromorelin's approval provides a foundation for exploring similar treatments in obese cats and dogs, particularly those with comorbid conditions that affect appetite and energy balance [33,112–114]. Research has explored the use of ghrelin receptor inverse agonists to induce inverse ghrelin actions, showing potential in reducing food intake, body weight, and improving glucose metabolism [115]. Additionally, agents targeting ghrelin directly, such as inhibitors of ghrelin O-acyltransferase, are being investigated as potential therapeutics for ghrelin-related diseases, including obesity and type 2 diabetes [116]. Furthermore, the development of ghrelin antagonists, including AG-blocking agents and novel GOAT inhibitors, presents a promising avenue for combating obesity by modulating ghrelin activity and its downstream metabolic effects [117]. While the therapeutic potential of ghrelin in obesity management is promising, careful consideration of dosages and delivery mechanisms is necessary to avoid overstimulation of appetite. Ongoing research and clinical trials is essential to optimize ghrelin-based therapies, ensuring they are safe and effective for use in companion animals.

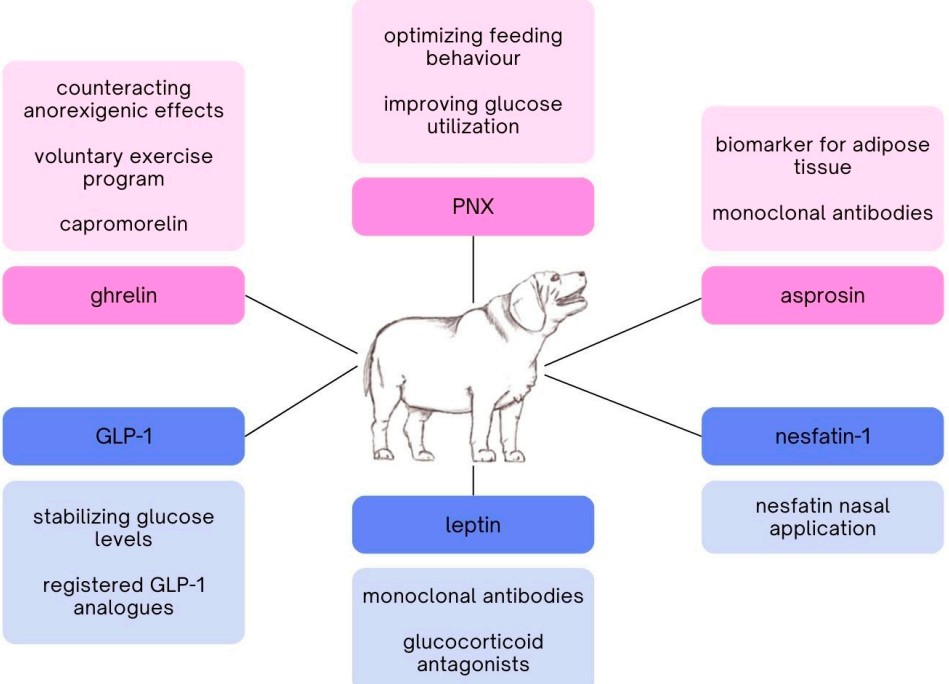

**Figure 2.** Therapeutic applications of orexigenic (pink boxes) and anorexigenic (blue boxes) peptides.

PNX's mechanism of action offers several avenues for therapeutic exploitation in the management of obesity, appetite dysregulation, and diabetes in cats and dogs (Figure 2). While research on the direct orexigenic effects of PNX in companion animals is limited and the specific effects of PNX in dogs and cats have not been extensively studied, its conservation across species implies a potential role in regulating food intake and other physiological processes in these animals [40,118]. Given that PNX can be delivered directly in vivo and

has shown the capacity to stimulate food intake dose-dependently in rodent models, it can be utilized to regulate and optimize feeding behavior in companion animals [119]. Additionally, PNX's impact on insulin sensitivity and glucose metabolism positions it as a candidate for addressing obesity-related metabolic disorders, such as diabetes, by improving glucose utilization and enhancing insulin sensitivity [118]. In practical applications, PNX-based therapies could involve the administration of PNX or its analogs to stimulate controlled food intake, thereby aiding in the management of obesity by regulating the meal size, meal duration, and overall food intake rate [40]. This approach not only targets weight management but also addresses secondary complications such as metabolic syndrome and fatty liver disease, as demonstrated in studies where PNX-14 administration significantly mitigated high-fat diet (HFD)-induced obesity and improved liver function in mice models [118]. Moreover, PNX's role in enhancing the expression of genes responsible for glycolysis and its influence on energy homeostasis further underscores its utility in treating obesity-related conditions. For instance, improving liver lipid metabolism and reducing systemic inflammation, which are critical in managing non-alcoholic fatty liver disease (NAFLD) associated with obesity, can be achieved through PNX-14 mediated pathways [118]. Thus, the development of pharmaceutical agents that mimic or enhance PNX's metabolic effects could provide a comprehensive approach to obesity management in veterinary patients, addressing both primary weight issues and associated metabolic complications [120]. Additionally, the therapeutic potential of PNX extends to mitigating obesity-induced cardiovascular conditions and behavioral issues, thereby improving the overall health and quality of life of obese companion animals [121]. The implementation of PNX-based therapies requires further research to elucidate its precise mechanisms of action, optimize dosage strategies, and ensure safety and efficacy in clinical settings, paving the way for innovative treatments in veterinary medicine [17].

The multifaceted role of asprosin in obesity and metabolic regulation presents intriguing therapeutic opportunities. Elevated circulating levels of asprosin have been consistently associated with obesity in both humans and animal models. Moreover, observational studies have highlighted its potential as a biomarker for adipose tissue mass, with higher asprosin levels correlating with increased body weight and adiposity (Figure 2). However, conflicting findings suggest a complex interplay between asprosin and obesity, necessitating further investigation to elucidate causative relationships. Mechanistically, asprosin exerts its effects through both peripheral and central pathways. Peripheral actions involve binding to the olfactory receptor on hepatocytes, stimulating hepatic glucose production via the cAMP signaling pathway. This glucogenic effect contributes to hyperglycemia and insulin resistance observed in obesity. Centrally, asprosin crosses the blood–brain barrier and activates appetite-stimulating neurons in the hypothalamic ARC via the G protein-cAMP-PKA pathway. By modulating feeding behavior, asprosin promotes excess energy intake and contributes to obesity development. Therapeutically, targeting asprosin signaling pathways presents a promising approach for managing obesity and associated metabolic disorders. Neutralization of circulating asprosin with monoclonal antibodies has been shown to reduce appetite and improve glycemic control in obese animal models, offering potential strategies for pharmacological intervention [122]. Additionally, anti-asprosin antibody therapy has been linked to decreased food intake, body weight, and glucose intolerance by inhibiting orexigenic hypothalamic neurons, including AgRP/NPY neurons, via stimulating small-conductance calcium-activated potassium currents [123]. Furthermore, small-molecule inhibitors of asprosin receptors, such as OR4M1, hold promise for oral dosing and combination therapy [122]. Elucidating the intricate mechanisms underlying asprosin's actions may uncover novel therapeutic targets for obesity treatment, addressing the growing global burden of the metabolic syndrome. In veterinary medicine, exploring the therapeutic potential of asprosin modulation in companion animals like cats and dogs could offer new avenues for managing obesity-related disorders. Given the similarities in metabolic regulation across species, insights gained from human and rodent studies may translate to veterinary applications. However, further research is needed to investigate

the safety, efficacy, and long-term effects of targeting asprosin pathways in companion animals. Overall, asprosin emerges as a compelling therapeutic target with the potential to revolutionize obesity treatment strategies and improve metabolic health across species.

Registered GLP-1 analogues (GLP-1As) used as medications for obesity include liraglutide and semaglutide (Figure 2) [124]. Interestingly, liraglutide has been studied in dogs with type 1 diabetes, yielding satisfactory results in stabilizing glucose levels [125]. The same GLP-1As was used in cats and resulted in weight loss. However, it is worth noting that this weight loss may have been caused by vomiting, which occurred in several of the cats studied [126]. GLP-1As treatment generally results in weight loss effects. However, during therapy, gastrointestinal side effects such as nausea are observed, which sometimes discourage patients from using these drugs [124,127]. The drugs are administered subcutaneously, so they do not have a single specific target site, which may lead to side effects. Additionally, although GLP-1As do not appear to pose a threat to health, their association with neoplasm development has been demonstrated in humans. Therefore, before introducing GLP-1As for obesity treatment in animals, the risk of carcinogenesis should be assessed [61]. Another long-acting GLP-1A, exenatide, primarily used in diabetes treatment, was studied in dogs and cats for weight loss. In the study with cats, a modified [Gln$^{28}$]exenatide was used, which, due to its prolonged action, is administered subcutaneously and only once a month, making it a very convenient solution, especially compared with exenatide administered subcutaneously twice a day in dogs. In both cases, a decrease in body weight was noted [128,129]. As it is already known, endogenous GLP-1 is almost immediately degraded by DPP4, which is also a T-cell stimulator. One could consider therapy aimed at inhibiting the activity of this enzyme and thereby prolonging the action of GLP-1 by mimicking its analogues. However, the associations of the enzyme with the immune system could lead to immune disturbances as a result of such therapy [61]. The newest solution proposed by Petersen et al. is the use of a bimodal molecule, integrating an NMDA receptor antagonist MK-801 with GLP-1 receptor agonism, effectively reversing obesity, hyperglycemia, and dyslipidemia in rodent models of metabolic disease. GLP-1A is linked with MK-801 through a chemically cleavable disulfide linker sensitive to redox, allowing for intracellular release of the NMDA receptor antagonist. The study suggests that this combination is more effective than using GLP-1A alone or MK-801 alone, and it also has a higher safety profile [130].

As previously mentioned in Leptin section, attempts to treat obesity with this hormone have failed due to leptin resistance occurring in many obese individuals. Leptin treatment only works in individuals with initially low leptin concentration and in the case of lipodystrophy [131]. However, an effective method of treating obesity with leptin has been described, which is somewhat the opposite of administering this hormone. This therapy is based on the assumption that the primary cause of impaired leptin action in leptin resistance may be the strong feedback mechanisms induced by the continuous activation of leptin signaling. Partial neutralization of leptin using monoclonal antibodies results in significant weight loss with simultaneous anti-diabetic effects (Figure 2). Monotherapy with leptin-neutralizing antibodies can be supplemented with other treatments, such as GLP-1 receptor agonists. It has been shown that GLP-1 combined with leptin exerts anorexigenic effects and enhances its central action. However, this therapy requires a low-lipid diet, during which substantial weight loss (over ~28%) is observed [74,132]. Interestingly, in the treatment of obesity and leptin resistance, or precisely its consequences, the endocannabinoid system may be helpful. Cannabinoid type 1 receptor (CB1R) inverse agonist increases adrenaline in WAT, promotes glomerular filtration, and boosts megalin expression, resulting in lower leptin levels in DIO mice and reversal of leptin resistance [133]. When it comes to ER stress inhibiting leptin signalling, a number of compounds have been proposed to alleviate ER stress and improve leptin responsiveness, such as fluvoxamine, flurbiprofen, and caffeine [76]. These compounds may seem like a good choice for humans, but in dogs and cats, caffeine and flurbiprofen (NLPZ, a popular medicine for sore throat) cause severe toxicity [134,135]. On the other hand, fluvoxamine, which belongs to selective serotonin

reuptake inhibitors (SSRIs), is safe for these species [136]. There are also reports indicating inhibition of leptin signaling in the hypothalamus by PTP1B and Tyrosine Phosphatase of T cells (TCPTP), contributing to obesity. To counteract this effect, Dodd et al. administered intranasal PTP1B inhibitor and RU486, a glucocorticoid antagonist, to mice daily and found that it resulted in reduced food intake and weight loss [137]. Intranasal drug administration in dogs and cats is an effective method, in some cases better than oral administration, more comfortable and safer [138,139].

As nesfatin-1 is a relatively new peptide and there is still a lack of information regarding its mechanisms of action, there are nonetheless studies suggesting it could be a potential treatment for obesity. In most studies, intraventricular administration of nesfatin-1 to mice and rats resulted in a reduced food intake [100,140]. Such a route of administration, however, is not a good solution for the therapeutic delivery of drugs. One study involved the intranasal application of nesfatin-1 in rats. Administering 10 nmoles of nesfatin-1 per rat resulted in appetite suppression of 6 h. As previously mentioned, the intranasal route of administration is highly effective in delivering the compound to the brain and, importantly, is feasible in animals (Figure 2). Moreover, a weight loss effect was observed with nesfatin-1 administration even in cases of leptin resistance [141]. Several studies indicate a relationship between the levels and actions of nesfatin-1 and sex hormones, so research is needed, stratified not only by sex but also by neutered and non-neutered individuals [142,143]. As the mechanisms of expression, action, and breakdown of nesfatin-1 are not well understood, at present, it is not possible to design another form of obesity therapy involving nesfatin-1 other than administering analogues of this compound.

## 6. Conclusions

Orexigenic and anorexigenic peptides offer promising therapeutic avenues for managing obesity and related metabolic disorders in companion animals such as cats and dogs. These peptides can help regulate appetite, improve feeding behaviors, and enhance metabolic health, contributing to more effective weight management. For instance, peptides like ghrelin and PNX can stimulate appetite, helping establish consistent feeding schedules and reducing binge eating. These peptides can also improve glucose metabolism and insulin sensitivity, aiding in the management of diabetes and other metabolic disorders. Additionally, ghrelin's influence on the central hedonic dopamine system can boost motivation for physical activity, making exercise more appealing and rewarding. This increase in activity levels promotes better weight management. Targeting peptides such as asprosin and using GLP-1 analogs can help manage hyperglycemia, insulin resistance, and other metabolic dysfunctions. These treatments can improve liver lipid metabolism, reduce systemic inflammation, and address conditions like non-alcoholic fatty liver disease. Innovative delivery methods, such as intranasal administration of peptides like nesfatin-1, offer practical and effective routes, enhancing therapeutic outcomes. Compared to subcutaneous administration, intranasal delivery is non-invasive and avoids the stress and discomfort associated with injections. This method ensures better compliance, particularly for long-term treatments. Intranasal delivery also provides a rapid route to the brain, where peptides can exert their effects more directly and efficiently, enhancing their anorexigenic or orexigenic properties. Moreover, intranasal administration can bypass first-pass metabolism in the liver, ensuring more of the peptide reaches its target sites in an active form. These advantages make intranasal administration a preferable option for delivering peptides like nesfatin-1 in veterinary applications. Combining peptide therapies with other treatments, such as monoclonal antibodies or small molecule inhibitors, can further enhance efficacy and safety.

However, challenges and considerations remain. Careful consideration of dosages and delivery mechanisms is necessary to avoid adverse effects such as overstimulation of appetite or gastrointestinal disturbances. Further research and clinical trials are essential to optimize these therapies, ensuring their safety and effectiveness in veterinary medicine.

Overall, integrating orexigenic and anorexigenic peptides into obesity treatment regimens for pets holds significant potential to improve their health and quality of life.

**Author Contributions:** Conceptualization, K.K. and C.O.-W.; methodology, K.K. and C.O.-W.; validation, M.B.A.; investigation, K.K. and C.O.-W.; writing—original draft preparation, C.O.-W. and K.K.; writing—review and editing, E.T., S.M. and M.B.A.; supervision, M.B.A. All authors have read and agreed to the published version of the manuscript.

**Funding:** This research received no external funding.

**Institutional Review Board Statement:** Not applicable.

**Data Availability Statement:** Data derived from public domain resources.

**Conflicts of Interest:** The authors declare no conflicts of interest.

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
