# Peer review of "Examining the Potential Applicability of Orexigenic and Anorexigenic Peptides in Veterinary Medicine for the Management of Obesity in Companion Animals"

_cimb, doi:10.3390/cimb46070401_

Round 1

Reviewer 1 Report

Comments and Suggestions for Authors

The review  article ‘Examining the Potential Applicability of Orexigenic and Anorexigenic Peptides in Veterinary Medicine for the Management  of Obesity in Companion Animals submitted by Osiak-Wicha et al.,  is  a very written and informative paper. The review is detailed and comprehensive and provides excellent information - not only on  the physiological mechanisms of these  peptides but also provides  a good overview of the therapeutic  applications of these biological agents.

My recommendation  prior to publication (associated with introduction/figure 1)- the authors identify and briefly expand information: 1) metabolic disorders/endocrinopathies (figure 1 -blue box)   that are associated with  obesity in various companions animals obesity. These could include hypothyroidism, hyperadrenocorticism diabetes mellitus  and so forth  2)  Briefly provides some information on the neoplasia that have been associated  with obesity  in companion animals  (figure 1 - yellow box). As examples mammary tumors and urogenital cancer and so forth.

Author Response

Reviewer #1

Dear Reviewer,

We would like to express our sincere gratitude to the Reviewer for their time, support, and kind comments. We have carefully read his/her suggestions and have made the necessary corrections that we hope will meet the Reviewer's approval. We have highlighted the revisions in the manuscript. In response to the Reviewer's comments, we have addressed each point as follows:

Comment 1. The review article ‘Examining the Potential Applicability of Orexigenic and Anorexigenic Peptides in Veterinary Medicine for the Management of Obesity in Companion Animals submitted by Osiak-Wicha et al.,  is  a very written and informative paper. The review is detailed and comprehensive and provides excellent information - not only on  the physiological mechanisms of these  peptides but also provides  a good overview of the therapeutic  applications of these biological agents.

Response: Thank you for your valuable feedback and kind words.

Comment 2. My recommendation prior to publication (associated with introduction/figure 1)- the authors identify and briefly expand information: 1) metabolic disorders/endocrinopathies (figure 1 -blue box) that are associated with  obesity in various companions animals obesity. These could include hypothyroidism, hyperadrenocorticism diabetes mellitus  and so forth  2)  Briefly provides some information on the neoplasia that have been associated  with obesity  in companion animals  (figure 1 - yellow box). As examples mammary tumors and urogenital cancer and so forth.

Response: Thank you for your comment, we have followed the suggestion and changes have been made to the manuscript and figure.

Reviewer 2 Report

Comments and Suggestions for Authors

Osiak-Wicha et al. offer a comprehensive examination of the role of peptides in the treatment of obesity in companion animals (cats and dogs). They emphasise the clinical potential of anorexigenic and orexigenic peptides in the regulation of appetite and energy balance, as well as the significance of molecular biology in the development of targeted therapies for companion animals. This is an intersesting and informative review. I have no further comments. 

Author Response

Reviewer #2

Dear Reviewer,

We would like to express our sincere gratitude to the Reviewer for their time, support, and kind comments.

Comment 1: Osiak-Wicha et al. offer a comprehensive examination of the role of peptides in the treatment of obesity in companion animals (cats and dogs). They emphasise the clinical potential of anorexigenic and orexigenic peptides in the regulation of appetite and energy balance, as well as the significance of molecular biology in the development of targeted therapies for companion animals. This is an intersesting and informative review. I have no further comments.

Response: Thank you for your kind words.